# NOD2 Agonism Counter-Regulates Human Type 2 T Cell Functions in Peripheral Blood Mononuclear Cell Cultures: Implications for Atopic Dermatitis

**DOI:** 10.3390/biom13020369

**Published:** 2023-02-15

**Authors:** Vladimir-Andrey Gimenez-Rivera, Harshita Patel, Franck P. Dupuy, Zoulfia Allakhverdi, Charlie Bouchard, Joaquín Madrenas, Robert Bissonnette, Ciriaco A. Piccirillo, Carolyn Jack

**Affiliations:** 1Infectious Diseases and Immunity in Global Health, Center for Translational Biology, The Research Institute of the McGill University Health Center, Montréal, QC H4A 3J1, Canada; 2Division of Experimental Medicine, Department of Medicine, McGill University, Montréal, QC H4A 3J1, Canada; 3IntegraSkin GmbH, 72076 Tübingen, Germany; 4Department of Medicine, David Geffen School of Medicine at UCLA, Los Angeles, CA 90095, USA; 5Innovaderm Research, Montréal, QC H2X 2V1, Canada; 6Department of Microbiology and Immunology, McGill University, Montréal, QC H3A 2B4, Canada; 7Division of Dermatology, Department of Medicine, McGill University, Montréal, QC H4A 3J1, Canada

**Keywords:** atopic dermatitis, thymic stromal lymphopoietin, interleukin-13, type 2 cytokines, T helper 2 and cytotoxic type 2 T cells, *Staphylococcus aureus*, *Staphylococcus aureus* enterotoxin B, nucleotide-binding oligomerization domain-containing protein 2, peptidoglycan, thymus and activation-related chemokine, signal transducer of activated T cells, GATA Binding Protein 3

## Abstract

Atopic dermatitis (AD) is known as a skin disease; however, T cell immunopathology found in blood is associated with its severity. Skin *Staphylococcus aureus* (*S. aureus*) and associated host–pathogen dynamics are important to chronic T helper 2 (Th2)-dominated inflammation in AD, yet they remain poorly understood. This study sought to investigate the effects of *S. aureus*-derived molecules and skin alarmins on human peripheral blood mononuclear cells, specifically testing Th2-type cells, cytokines, and chemokines known to be associated with AD. We first show that six significantly elevated Th2-related chemokine biomarkers distinguish blood from adult AD patients compared to healthy controls ex vivo; in addition, TARC/CCL17, LDH, and PDGF-AA/AB correlated significantly with disease severity. We then demonstrate that these robust AD-associated biomarkers, as well as associated type 2 T cell functions, are readily reproduced from healthy blood mononuclear cells exposed to the alarmin TSLP and the *S. aureus* superantigen SEB in a human in vitro model, including IL-13, IL-5, and TARC secretion as well as OX-40-expressing activated memory T cells. We further show that the agonism of nucleotide-binding oligomerization domain-containing protein (NOD)2 inhibits this IL-13 secretion and memory Th2 and Tc2 cell functional activation while inducing significantly increased pSTAT3 and IL-6, both critical for Th17 cell responses. These findings identify NOD2 as a potential regulator of type 2 immune responses in humans and highlight its role as an endogenous inhibitor of pathogenic IL-13 that may open avenues for its therapeutic targeting in AD.

## 1. Introduction

Atopic dermatitis (AD) is the leading cause of disability of all skin diseases, affecting ~20% of children globally [1,2,3]. Once considered a disease of childhood, AD is found prevalent, unresolving, and burdensome across the lifespan, often manifesting with severe, treatment-resistant chronic Th2 inflammation in adults [4,5,6].

Recent studies have shown that Th2 cells require epithelial signaling at sites of inflammation before achieving full effector function or reactivation [7,8]. Keratinocytes respond to skin damage and/or infection by producing “alarmin” cytokines: IL-25 (IL-17E), IL-33, and thymic stromal lymphopoietin (TSLP) [9,10]. These alarmins prime, polarize, and recruit type 2 responses; can arrest Th1/Th17 development; and induce type 2 co-stimulatory receptors, such as OX40L on dendritic cells [11]. Alarmins are thus considered end-organ checkpoints for epithelial type 2 immune responses [12]. TSLP is highly expressed in AD skin [12] and is triggered from keratinocytes in response to injury and/or to *Staphylococcus aureus* (*S. aureus*) [13]. TSLP can even act directly on primed CD4 T cells; indeed, AD patients have elevated TSLP-receptor (TSLPR^+^) T cells, which correlate with disease severity [14,15,16]. Skin-derived TSLP may in fact be sufficient to support the atopic march from skin sensitization to asthma [17,18,19,20,21]; yet, since TSLP also has homeostatic roles, co-factors in the surrounding microenvironment are likely important. AD is complicated by strong environmental influences, whereby *S. aureus* is a key factor, colonizing ~75–90% of skin lesions [22,23,24,25,26]. Emerging evidence supports an adjuvant capacity for this pathogen to impair tolerance and drive allergic sensitization and inflammation in atopic disorders [27], yet debate persists on its bystander versus causal role in AD. *S. aureus* can drive inflammation via numerous virulence factors, whereby superantigens are most prominently associated, especially staphylococcal enterotoxin B (SEB) [28,29,30,31,32,33]. The severity of AD is well-known to be associated with sensitization to SEB [34]. *S. aureus* is also known for numerous immuno-evasive mechanisms that limit effective immune responses, promoting its own survival and impeding proper immunity in humans [35,36]; as such, studies are needed to clarify such contrasting immune effects related to AD. 

Protective host skin immunity to *S. aureus* is complex but involves innate signaling and IL-17 family cytokines, critical to recruiting neutrophils to epithelia, thus limiting *S. aureus* growth, and protecting the host from deep tissue invasion and infection [37,38]. As a facultative intracellular anaerobe, immunity to intracellular *S. aureus* also requires cell-mediated Th1 (IFNγ) responses [37]. Normal human skin has abundant *S. aureus*-specific tissue-resident memory CD4+ T cells, producing Th17 (IL-17A) and Th1 (IFNγ) responses [39]. In diseased AD skin, such responses are either missing, ineffective, or evaded by *S. aureus* [37]; instead, pathogenic Th2 (IL-13) and related cellular responses central to AD consistently prevail, which may in fact facilitate survival of *S. aureus*. Seminal work by Biedermann (2010) demonstrated that murine DCs coactivated by *S. aureus* cell-wall peptidoglycan (PGN) monomers act to suppress Th2 cell priming and generate protective Th1 and Th17 responses [40]. The cognate receptor, nucleotide-binding oligomerization domain-containing protein (NOD)2, is now known to be essential for clearing *S. aureus* infections in mice [41,42]. NOD2 is expressed by human antigen-presenting cells (APCs), such as dendritic cells (DCs), as well as keratinocytes where agonism drives IL-17C in response to *S. aureus* [43]. Notably, Th2 and Th17 cells have been shown to be reciprocally regulated; yet, such missing ‘effective’ *S. aureus* Th17 responses remain understudied in AD [44]. Understanding mechanisms driving dysfunctional Th2 effector functions despite *S. aureus* skin burden is critical to AD patients; we thus sought to determine the effect of NOD2 agonism in human peripheral blood mononuclear cells and its downstream impact on type 2 T cell functions.

The aim of our study was to establish an in vitro model based on skin-derived factors relevant to AD that can replicate type 2 T-cell and pathogenic IL-13 cytokine functions in disease. Using this human model, we sought to investigate the immunomodulatory effects of key innate epithelial *S. aureus* sensor NOD2, specifically its capacity to regulate type 2 T cell AD-like pathology.

## 2. Materials and Methods

### 2.1. Study Design

Adult atopic dermatitis patients and healthy subjects consented in writing to the McGill Dermatitis Database and Biobank (REB 2020-5565) and appended case-control “Cross-talk” (REB 2017-2571) for clinical data and blood analyses. Eligibility criteria for patients included adult age, meeting Hanifin & Rajka diagnostic criteria, and moderate to very severe disease as per the Eczema Area and Severity Index (EASI); median 23.6, n = 15, absence of systemic therapy according to published wash-out periods, which align with larger cohorts found in clinical trials [45]. Age-matched healthy subjects were enrolled if exclusion criteria were met, including a lack of atopic disease and personal and/or family history (n = 7). Blood samples were collected from patients and healthy subjects. Physician reported clinical parameters (EASI, vIGA, BSA) were collected from participants. Clinical laboratory tests were used to obtain IgE levels from AD patients (normal, <200 kU/L). 

### 2.2. Isolation, Freezing, and Thawing of Peripheral Blood Mononuclear Cells 

Peripheral blood mononuclear cells (PBMC) were isolated from whole blood using Ficoll-Paque Plus (Cytiva Life Sciences, Marlborough, MA, USA) gradient centrifugation. Red blood cell lysis was performed with eBioscience 1X RBC Lysis Buffer (Invitrogen/Thermo Fisher Scientific, Waltham, MA, USA). To freeze PBMCs following isolation, 10 mln cells/mL were resuspended in heat-inactivated fetal bovine serum with 10% dimethyl sulfoxide (DMSO) (Sigma-Aldrich, St. Louis, MO, USA) in cryogenic vials. PBMCs were thawed in a 37 °C water bath and washed with pre-warmed complete RPMI 1640 (cRPMI) with L-glutamine (Gibco/Thermo Fisher Scientific, Waltham, MA, USA) supplemented with 10% FBS, and 1% of sodium pyruvate, hepes, penicillin/streptomycin and non-essential amino acids (each, Wisent, Saint-Jean-Baptiste, Waltham, MA, USA). Cells were washed with cRPMI and used for multiparametric flow cytometry staining.

### 2.3. In Vitro Peripheral Blood Mononuclear Cell (PBMC) Model 

Freshly isolated PBMCs were resuspended in complete RPMI 1640 (Gibco) supplemented with 10% autologous human plasma, 1 mM sodium pyruvate (HyClone, Logan, UT, USA), 25 mM hepes, 100 μg/mL penicillin/streptomycin (HyClone), 2mM L-glutamine, and 1X non-essential amino acids. In a 24-well plate coated with collagen I from rat tail (Corning, Corning, NY, USA), 2.5 × 10^6^ cells/well in 450 μL were seeded in triplicates and rested overnight at 37 °C in a 5% CO_2_ incubator. To replicate the inflammatory milieu of AD skin, PBMCs were stimulated with 2 ng/mL of staphylococcal enterotoxin B (SEB) (Toxin Technologies, Sarasota, FL, USA) and 120 ng/mL of recombinant human thymic stromal lymphopoietin (TSLP) (Biolegend, San Diego, CA, USA) at Day 0. Every 2 days, complete RPMI was added to each well. On day 7, the cells were harvested and cell count for each condition was performed. In 96-well V bottom plates, 2.5 × 10^5^ cells/well were plated for subsequent flow cytometry experiments.

### 2.4. Functional Assays 

Commercial and clinical *S. aureus* strains and fractions tested were previously described [46,47]. For NOD2 agonism, ultrapurified *S. aureus* peptidoglycan (PGN-SAndi, Invivogen) was used, abbreviated as NOD2L. Cells were seeded in 96-well plates (2.5 × 10^5^ cells) and stimulated as indicated. When NOD2 agonist used, cells were incubated for 1h prior to stimulation, using vehicle as a control; cells re-stimulated using Thermo Fisher Scientific T-activator CD3/CD28 Dynabeads at a 1-to-1 ratio. Cell-free supernatants were collected and stored at −20 °C until analyzed.

### 2.5. Extracellular and Intracellular Flow Cytometry Staining 

For intracellular cytokine detection, cells were stimulated with 20 ng/mL of phorbol 12-myristate 13 acetate (PMA) (Sigma-Aldrich) and 750 ng/mL of ionomycin (Sigma-Aldrich) in the presence of Monensin based Golgi Stop (1:1000 dilution) for 3 h at 37 °C in 5% CO_2_ incubator prior to staining. Single cell suspensions were stained with Fixable Viability Dye eFluor 780 (eBioscience, San Diego, CA, USA, 1:1000) and Human TruStain FcX blocking solution (Biolegend, 1:100) in PBS (Wisent) for 15 min at 4 °C. Extracellular surface markers were stained for 30 min at 4 °C with the following fluorochrome-conjugated monoclonal antibodies prepared in PBS (according to experiment): anti-CD4 FITC (RPA-T4, BD Biosciences, Mississauga, ON, Canada), anti-CD4 APC (RPA-T4, Biolegend), anti-CD8𝛼 PerCP-Cy 5.5 (RPA-T8, BD Biosciences), anti-CD8𝛼 V500 (RPA-T8, BD Biosciences), anti-CD3 BV785 (OKT3, Biolegend), anti-CD45RA AF700 (HI100, Biolegend), and anti-HLA-DR V500 (G46-46, BD Biosciences). Subsequently, cells were fixed and permeabilized using the Foxp3 Transcription Factor Fixation/Permeabilization buffer set (eBioscience) (ex vivo) or True-Nuclear Transcription Factor Fixation/Permeabilization Buffer set (Biolegend) (in vitro cultures). Intracellular staining was performed with fluorochrome-conjugated monoclonal antibodies anti-GATA3 BUV395 (L50-823, BD Biosciences), anti-IL-13 PE (JES10-5A2, BD Biosciences), anti-IL4 APC (8D4-8, BD Biosciences), anti-IFN-y PE-Cy7 (4S.B3, BD Biosciences), anti-IL17a V450 (N49-653, BD Biosciences), and anti-IL-2 PerCP-Cy 5.5 (MQ1-17H12, BD Biosciences). Cells stimulated with PMA and ionomycin were stained intracellularly for CD4 and CD8𝛼. Samples were acquired on the LSRFortessa X-20 flow cytometer (BD Bioscience) and analyzed using FlowJo version 10 software (Tree Star, Ashland, TN, USA). For the IL-13 secretion assay, after 6h re-stimulation, cells were stained as per protocol (Miltenyi, Gaithersburg, MD, USA, 130-093-479). 

### 2.6. Phosphoflow Nuclear Staining

For intracellular phosphorylated protein detection, cells were re-stimulated with human anti-CD3/CD28/CD2 (StemCell, Vancouver, Canada, 1:100) for 6 or 24 h at 37 °C in 5% CO_2_ incubator prior to staining. Nuclear permeabilization was performed with BD Phosflow™ Perm Buffer III (BD Biosciences) on ice for 30 min. Cells were then washed twice with PBS. Extracellular and intranuclear markers were stained at room temperature for 30 min with the following monoclonal antibodies raised against human proteins prepared in permeabilization buffer, Human TruStain FcX blocking solution (Biolegend), and Brilliant Stain buffer (BD Biosciences): anti-CD4 APC (RPA-T4, Biolegend), anti-CD8𝛼 BV605 (SK1, Biolegend), anti-CD3 V500 (UCHT1, BD Biosciences), anti-CD45RO APC/Fire 750 (UCHT1, Biolegend), anti-GATA3 PE (L50-823, BD Biosciences), anti-OX40 BV711 (Ber-ACT35, Biolegend), anti-ICOS AF700 (C398.4A, Biolegend), and anti-HLA-DR BV395 (G46-6, BD Biosciences). Cells were washed and resuspended in PBS for acquisition. For some experiments cells were acquired with Amnis^®^ ImageStream^®^XMark II system and analyzed with IDEAS^®^ Version 6.2 (Image Data Exploration and Analysis Software) following the nuclear colocalization method pre-installed in this software.

### 2.7. Multiplex Cytokine/Chemokine Array Immunoassays 

Ex vivo plasma fractions and in vitro model supernatants were analyzed using Luminex xMAP technology for multiplexed quantification of 71 Human cytokines, chemokines, and growth factors. Multiplexing analysis was performed using the Luminex™ 200 system (Luminex, Austin, TX, USA) by Eve Technologies (Calgary, AB, Canada). Seventy-one markers were simultaneously measured in the samples using Human Cytokine 71-Plex Discovery Assay^®^, which consists of one 48-plex and one 23-plex (MilliporeSigma, Burlington, MA, USA), according to the manufacturer’s protocol. The 48-plex consisted of sCD40L, EGF, Eotaxin, FGF-2, FLT-3 Ligand, Fractalkine, G-CSF, GM-CSF, GROα, IFN-α2, IFNγ, IL-1α, IL-1β, IL-1RA, IL-2, IL-3, IL-4, IL-5, IL-6, IL-7, IL-8, IL-9, IL-10, IL-12(p40), IL-12(p70), IL-13, IL-15, IL-17A, IL-17E/IL-25, IL-17F, IL-18, IL-22, IL-27, IP-10, MCP-1, MCP-3, M-CSF, MDC, MIG/CXCL9, MIP-1α, MIP-1β, PDGF-AA, PDGF-AB/BB, RANTES, TGFα, TNF-α, TNF-β, and VEGF-A. The 23-plex consisted of 6CKine, BCA-1, CTACK, ENA-78, Eotaxin-2, Eotaxin-3, I-309, IL-16, IL-20, IL-21, IL-23, IL-28A, IL-33, LIF, MCP-2, MCP-4, MIP-1δ, SCF, SDF-1α+β, TARC, TPO, TRAIL, and TSLP. Assay sensitivities of these markers range from 0.14–55.8 pg/mL for the 71-plex. Individual analyte sensitivity values are available in the MILLIPLEX^®^ MAP. Additional (focused) analyses were performed with in vitro model supernatants using the T helper cytokine panel (13 plex) (Biolegend). Briefly, supernatants were incubated with variable size beads that captured each specific cytokine; bead-cytokine complexes were then incubated with fluorescent-labeled cytokine-specific detection antibodies. Mean fluorescence intensity (MFI) and FSC-A (beads size) were used to quantify each cytokine’s concentration using flow cytometry.

### 2.8. Statistical Analysis

Statistical testing varied according to the experimental design and is specified in the figure legends. Briefly, when normality could not be assumed in exploratory multiplex immuno-assay, the non-parametric Mann-Whitney test was used to compare the two unpaired groups, and *p* values were adjusted for multiple comparisons using Holm-Šídák’s test with an alpha 0.05. In sub-analyses of target cytokines with Gaussian distribution, the two groups were compared using un-paired *t*-tests per row, again followed by Holm-Šídák’s correction for multiple comparisons. For in vitro work with matched data sets, paired *t*-tests per row were followed by Holm-Šídák’s multiple comparisons test; for non-parametric data, Wilcoxon matched pairs signed-rank test was used. For comparison of three groups defined by one factor, analyses were performed with a one-way ANOVA, followed by Dunnett’s multiple comparison post hoc test. When testing multiple cytokine levels in ex vivo analyses, a one-way ANOVA was calculated for each cytokine and corrected with Dunnett’s multiple comparisons test with *p*-values as in the post hoc test. All experiments were performed in duplicate or triplicate. A *p* value of ≤0.05 was considered significant. The *p* values were indicated as following: * *p*-values < 0.05; ** < 0.01; *** < 0.001; **** < 0.0001 in the figure legends. All statistical analyses were performed using GraphPad Prism, version 7 (GraphPad Software).

## 3. Results

### 3.1. Type 2 Inflammatory Chemokine Biomarkers Readily Distinguish Adult Atopic Dermatitis Patient Blood from Healthy Subjects 

To test and validate the detection of AD biomarkers, we characterized a small cohort of adult patients meeting diagnostic and severity criteria for moderate-to-very severe AD, compared to matched healthy subjects (see Appendix A for clinical characteristics), which align with larger cohorts found in clinical trials [45].

Using an extended multiplexed immunoassay for 71 cytokine/chemokine/growth factors in plasma, we detected significant differences in six Th2-related chemokines between AD patients and healthy subjects, as shown in Figure 1A. Five of these six chemokines are validated [48] as AD disease biomarkers: thymus and activation-regulated chemokine (TARC/CCL17), macrophage derived chemokine (MDC/CCL22), eosinophil-attracting chemokine (eotaxin-3/CCL26), cutaneous T cell attracting chemokine (C-TACK/CCL27), and monocyte chemoattractant protein-4 (MCP-4/CCL13). Interestingly, we also detected the significant elevation of I-309/CCL1, a chemoattractant for monocytes/macrophages and Th2 cells into inflammatory sites, known to be increased in AD skin lesions [49]. These chemokine biomarkers were found to correlate significantly with each other, as shown in Figure 1B and Appendix A. TARC correlated highly significantly with EASI disease severity (Figure 1C,D), as did the clinical laboratory test lactate dehydrogenase and platelet derived growth factor, PDGF-AA/BB, known for its effects on fibroblasts and for mediating airway inflammation and remodeling in asthma [50]. EASI correlation with other biomarkers (IgE, MDC, eotaxin-3, sCD40L, IL-9, and PDGF-A) were not significant, although trends were found (Figure 1C). Additional significantly elevated chemokines/cytokines/growth factors (BCA-1, VEGF-A, TNFα, MCP-3, IP-10, IL-13, FGF-2, sCD40L (all, *p* ≤ 0.05) and IL-5, and IL-9 (both *p* ≤ 0.01) did not meet the threshold following adjustments for multiple comparisons. We thus found robust biomarkers in the blood of adult AD patients, including the disease severity correlate TARC, and next sought to determine upstream signals capable of inducing these factors.

### 3.2. Human In Vitro Model Reproduces Type 2 Inflammatory Features of AD

To study etiologic factors relevant to adult AD disease, we established a simple in vitro PBMC-derived model, as illustrated in Figure 2A. Skin tissue-derived signals may be key to the Th2 cell pathology associated with AD; in particular, there is strong evidence for both TSLP and the *S. aureus*-derived superantigen SEB in disease [51]. We thus exposed PBMCs from healthy subjects to TSLP and SEB in culture and tested for a core set of internationally validated [48] AD biomarkers after one week, compared with those found ex vivo in AD patients (Figure 2B,C). We found that TARC/CCL17, MDC/CCL22, and MCP-4/CCL-13 were significantly elevated in the culture following a single exposure to TSLP/SEB, mimicking biomarkers found in AD blood ex vivo. Two of these Th2-related chemokines are produced by epithelia (CTACK/CCL27) or endothelium (eotaxin-3); as these cell types are not found in PBMCs, they were not expected to be increased. We next assessed the cytokine biomarker IL-13 in comparison to other master Th cytokines and found that TSLP/SEB significantly upregulated the Th2 cytokines IL-5, IL-9, and IL-13 in PBMCs from healthy subjects, replicating AD-like functional deviation (Figure 3). The Th2 cytokine IL-4 was very low in culture, mimicking the IL-13-dominated profile of AD [52]. No other effector Th subset cytokine (Th1, IFNγ; Th22, IL-22; or Th17, IL-17A) was significantly upregulated, nor was IL-6 or the regulatory cytokine IL-10 (data not shown). Comparable findings were generated using blood from AD patients in vitro (data not shown). Taken together, a singular exposure to TSLP/SEB is sufficient to drive PBMCs to produce AD biomarkers like TARC, known to correlate with disease severity, as well as master Th2 cytokines, without the contribution of epithelial cells nor exogenous cytokines, such as IL-4 or IL-2, nor artefactual Th1-blocking antibodies, as used in traditional in vitro Th2 models.

To examine TSLP/SEB-mediated AD-like functional Th2 deviation at the cellular level, we next utilized multiparametric flow cytometry to study immunophenotype after stimulation by staining for Th1/Th2 intracellular cytokines, as shown in Figure 4. Using PBMCs from healthy subjects, we found that TSLP/SEB led to significantly increased IL-13^+^ CD4 (Th2) and CD8 (Tc2) T cells (Figure 4A), but not IFNγ^+^ Th1 or Tc1 cells (Figure 4B). In AD patients, Tc2 cells were not increased following in vitro stimulation. Notably, TSLP/SEB also induced significant elevation in the Th2-related co-stimulatory receptors ICOS and OX40, as well as the master Th2 transcription factor GATA3, as shown in Figure 5. TSLP/SEB thus induced Th2/Tc2 intracellular cytokine, transcription factor, and co-stimulatory receptor expression in T cells.

### 3.3. Functional Modulation of Pathogenic Type 2 Cells 

Prior work by our group and others has demonstrated that certain *S. aureus* strains and cell wall products can induce immunosuppressive responses and/or Th2 inhibition [46]. Having established skin-derived factors mediating the induction of AD-related biomarkers, chemokines, and type 2 T cells, we sought to investigate counterregulatory *S. aureus*-mediated signals known to inhibit Th2 responses [40] using our in vitro model. 

#### 3.3.1. NOD2 Agonism in PBMCs Inhibits TSLP/SEB-Induced Type 2 Cytokine Secretion

We specifically focused on NOD2 agonism as PGN is a major constituent of the cell wall and chemical PGN alterations (blocking NOD2 recognition) feature as a prominent adaptive survival mechanism for *S. aureus*. We have previously found that TNFα production in response to various *S. aureus* strains was dependent on phagosome processing [47], and NOD2 is known to co-localize with phagosomes in innate immune cells [53]; therefore, we aimed to investigate a role for NOD2 signaling (or absence thereof) role in AD.

We tested our in vitro model following priming with a NOD2 agonist and found that memory CD4 and CD8 T cell IL-13 production was significantly inhibited relative to the control TSLP/SEB alone, as shown in Figure 6A. Culture supernatants were further tested for polar T cell cytokines induced following re-stimulation at 6 hrs. We found that PBMC cultured for 6 days in the presence of TSLP/SEB alone increased IL-4, IL-5, IL-13, IL-9, IL-10, TNF, as well as IL-22 and IL-6 production, as compared to untreated cells, following mitogen restimulation. Notably, adding the NOD2L significantly reduced Th2 (IL-4, IL-13, IL-5) and IL-10 cytokines relative to TSLP/SEB alone (Figure 6B). Interestingly, we found that NOD2 ligation significantly increased IL-6 secretion, along with smaller but significant increases in TNF and IFNγ. We also noted the lack of IL-10 induction in contrast to prior reports with cell wall extracts of *S. aureus* or TLR2 ligation [54,55]. We validated that similar inhibition could be replicated in PBMCs isolated from AD patients (data not shown). In summary, we found NOD2 ligation effectively counter-regulated TSLP/SEB-induced type 2 cytokines.

#### 3.3.2. NOD2 Agonism in PBMCs Induces Th17-Related Transcription Factor STAT3 

As NOD2 ligation significantly increased IL-6, known to be essential for Th17 development, we sought to determine if NOD2 ligation in PBMCs affected T cell transcription factors, specifically STAT3, which transduces IL-6 signaling downstream in Th17-cells. We found NOD2 agonism significantly increased phosphorylated STAT3 (pSTAT3), key to Th17 differentiation, in both CD4 and CD8 T cells in cultures at six hours following the re-stimulation (Figure 6C, left panel) and nuclear localization of pSTAT3 but not pSTAT5; when analyzed using Imagestream (Figure 6C, right panel), the latter was a molecule previously reported to induce Th2 differentiation downstream of TSLPR. We also found increased levels of Lck in CD4 T cells, a tyrosine kinase and CD4 co-factor known for higher expression in Th1 rather than in Th2 cells (data not shown) [56], but no significant difference in the aryl hydrocarbon receptor (AHR), STAT1, or JAK1 at this time point. In addition, we tested the effect of NOD2L on the master Th2 transcription factor GATA3 with PBMCs from AD patients. We found a trend for reduced expression of GATA3 phosphorylation in CD4 and CD8 T cells (data not shown). While we were not powered to study donor-to-donor heterogeneity, we noted that PBMCs from AD patients responded less frequently to NOD2L-mediated inhibition than healthy controls (HC: 86% and AD: 67%; data not shown). These results suggest that the NOD2 inhibition of type 2 T cell from PBMCs cultured in the presence of TSLP and SEB occurs via activation of counter-regulatory signaling pathways, leading to decreased IL-13 and increased IL-6 and pSTAT3, both critical to Th2 and Th17 responses, respectively.

## 4. Discussion

The novelty of our study is two-fold. First, we demonstrate that TSLP and SEB are sufficient to drive IL-13 secretion and pathogenic effector type 2 T cell responses from healthy PBMCs in vitro, replicating AD-like cytokine, chemokine, biomarker, and T cell functions found in patients ex vivo. Second, using this model, we showed that NOD2 ligation inhibits TSLP/SEB-mediated induction of type 2 cytokines and associated Th2 and Tc2 cell functions. These findings highlight cutaneous alarmin and *S. aureus* signals relevant to AD inflammation and identify NOD2 as a potential therapeutic target. 

While TSLP/SEB have previously been shown to increase CCL17 and IL-2 production in CD1c^+^ DCs, their impact on T cell function was not previously tested [57]. In 2002, Soumelis et al. reported that TLSP alone can drive Th2 differentiation from naive T cells, but only in an allogeneic system with purified CD11c^+^ DCs [12]. Watanabe et al. then demonstrated that (IL-4^+^) Th2 differentiation occurs only under artefactual blockade (anti-IL-12) of the Th1 pathway in autologous TSLP-DC-T cell cultures; without it, homeostatic memory T cells are produced [58]. Interestingly, in this study, TSLP-DCs that pulsed with SEB were shown to be activated, but the downstream effect on Th2 effector responses was not explored [58]. SEB-activation alone induces mixed (Th1,2,17), Th1-dominated responses in epithelial-T cell (skin-homing) co-cultures [51]. As such, the significance of our work is the generation of a novel, agile human model to study pathogenic type 2 T cell responses in an autologous system without foreign antigens, artefactual Th1 blockade, or costly isolation of purified DCs. In summary, a single exposure to TSLP/SEB is sufficient to drive immune hallmarks of AD, including type 2 chemokine biomarkers and pathogenic effector type 2 T cells secreting high IL-13 and IL-5 from healthy PBMCs, validated here to correlate with AD activity in adult patients with the moderate-to-severe disease [48]. 

After one week, TSLP/SEB-stimulated blood mononuclear cultures secreted the canonical Th2 cytokines IL-13, IL-5, and IL-9, but very limited IL-4. Interestingly, protein levels of IL-13, but not IL-4, are consistently detected and shown to be increased in AD skin across studies [52,59]. These findings suggest that the combination of skin derived TSLP/SEB is sufficient to generate canonical pathogenic AD cytokines in blood (Figure 3B). Stimulation in vitro significantly elevated IL-13^+^ Th2 and Tc2 cells and increased master transcription factor GATA3 as well as the key Th2-promoting co-stimulatory receptors ICOS and OX-40, efficiently replicating cellular profiles seen in AD patients. Our findings also add to existing evidence that OX40L on TSLP-activated DCs triggers Th2 deviation in the absence of IL-12 [60]. Interestingly, several monoclonal antibodies blocking the OX40L:OX40 axis have shown efficacy in AD and are currently in advanced human trials [61]. In our model, key AD biomarkers are reproduced, including TARC/CCL17, MDC/CCL22, and MCP-4/CCL-13; of note, biomarkers derived from epithelia or the endothelium (C-TACK/CCL27 and Eotaxin-3/CCL26) were not reproduced in our model due to a lack of endothelium/epithelium.

As introduced, pathology in AD is intricately linked to *S. aureus* with severity associated with sensitization to SEB [34]; yet, *S. aureus* is also known for numerous immuno-evasive mechanisms [35,36]. The contrasting immunomodulatory effects of *S. aureus* related to AD remain to be clarified. Healthy individuals have abundant *S. aureus*-specific tissue-resident Th17 (IL-17A) and Th1 (IFNγ) [39], consistent with protective cutaneous immune responses to this pathogen. Similarly, prior work has shown that *S. aureus* cell-wall derived PGN-derived monomers can inhibit Th2 responses in favor of Th1/Th17 responses in murine models, following NOD2/TLR2 agonism [18]. In humans, NOD2 activation has been shown in dendritic cells (DCs) to promote the development of Th17 cells and in keratinocytes to induce IL-17C [62]. Th17 cell function at epithelial surfaces is critical to clearing *S. aureus* via neutrophil recruitment [37]. In contrast, Th2 cell responses in AD may be conducive to survival of *S. aureus*, thus leading to positive microbial feedback via increased skin alarmins triggered by this virulent pathogen. While TLR2-mediated signaling is well defined, there are large gaps in understanding the wide range of NOD2-mediated immune responses in humans. NOD2 is recognized for its key role at epithelial surfaces with high microbial loads, such as the gut and the skin, co-localizing with phagosomes and activating autophagy [63,64]. NOD2 is thus poised to respond to the high burden of *S. aureus* peptidoglycan in AD and should drive protective antibacterial innate immune responses [53,65,66,67,68]. Given that a central facet of *S. aureus* virulence and manipulation of immune responses is its capacity to co-opt phagocytic programs, we postulate that AD pathobiology may involve altered phagosomal NOD2 activation by *S. aureus.* Thus, we sought to clarify NOD2-mediated modulation of type 2 T cell responses, more precisely whether innate NOD2 signaling may have the potential to counteract AD-like pathobiology.

We demonstrate here that the addition of a NOD2 ligand during in vitro type 2 T cell deviation restricts TSLP/SEB-mediated induction of type 2 cytokines and T cell functions, including the inhibition of the master AD cytokine IL-13, along with IL-4 and IL-5 secretion (Figure 6). NOD2 agonism increased IL-6 and pSTAT3 in healthy PBMCs, both critical regulators of the differentiation and function of Th17 cells [69]. Our data suggest that NOD2 activates innate myeloid cells (monocyte/macrophages and DCs) in our PBMC model, driving IL-6 secretion and downstream pSTAT3 activation in T cells. Although prior reports in a murine model indicated that NOD2 synergizes with TLR signaling to boost Th1 and Th17 responses and to suppress Th2 cell responses via DC-produced IL-10 [54,55], to the best of our knowledge, we are the first to demonstrate that NOD2 signaling in human PBMCs inhibits type 2 memory T cell expansion without IL-10 or exogenous TLR ligands. Notably, recent findings have shown that high IL-6 responses to heat-killed skin bacteria by human innate lymphoid cells is also NOD2-dependent [70,71]. The clinical relevance of the Th17-Th2 counter-regulatory mechanism is also highlighted by the monogenic Hyper-IgE syndrome, whereby defective STAT3 leads to the cardinal features of AD, including eczematous lesions on skin, *S. aureus* infections, and type 2 immune responses with high IgE [72]. 

Our findings are supported by numerous human studies demonstrating NOD2-driven production of IL-6 [73,74,75,76], TNF and IL-17 [76,77] in PBMCs, and IL-17 in memory Th cells exposed to NOD2-activated human DCs [62]. Additional NOD2-mediated pathways can also contribute to the regulation of type 2 T and innate lymphoid immune responses and require further study. Synthetic NOD2 agonists have been shown to enhance antigen presentation with DCs, stimulating T cell activation and proliferation, which may influence TCR signal strength and/or metabolic reprogramming important for Th2 effector programs [11,76]. NOD2 agonism can also upregulate IFNγ [78] in activated PBMCs and in PMBC-epithelial co-cultures [79], consistent with our findings. NOD2 is known to bind the scaffold protein receptor-interacting serine/threonine-protein kinase 2 (RIPK2), driving additional innate pathways, including the production of antiviral type 1 interferons [80]. These pathways must be further investigated, along with the role for NOD2 in mediating autophagy and/or wound healing responses within epithelial barriers that are still being defined. Atopic patients are also frequently polysensitized to protease allergens, such as house dust mites (HDM), which can alternatively activate PRRs, activate alarmins, and stimulate Th2 responses [81,82,83,84], a mechanism mimicked by murine ‘AD’ models, eliciting clonal responses to protein allergens. These responses contrast with polyclonal T cell repertoires in AD, as modeled here with SEB superantigen; thus, NOD2-mediated regulation of clonal allergen-specific T cells may differ. Interestingly, a NOD1 ligand was shown to inhibit allergen-induced airway inflammation in mice [85]. In summary, human-centric studies will be required to dissect species-specific and contextual NOD2 mechanistic pathways and their impact on human Th2 functional responses and atopic disorders in general. 

NOD1 and NOD2 polymorphisms have been identified in AD [86,87,88], and these receptors are interconnected, emphasizing the need for more expansive future studies [89]. *S. aureus* is also recognized to have its own mechanisms to impair effective host PGN responses. PGN is critical for bacterial viability and a key target for host lysozyme digestion by hydrolysis. *S. aureus* can increase its pathogenicity by chemically altering (O-acetylating) its PGN such that lysosomal digestion is blocked, leading to impaired host responses [90]. Recent work has shown that a lack of protective immunity to *S. aureus* in humans involves this PGN O-acetylation, thus limiting Th17 polarization [91]. Further studies are needed to determine if this *S. aureus* evasion strategy plays a role in the host–pathogen dynamics with *S. aureus* in AD. If so, there may be important opportunities for drug discovery, including a wide range of vaccines and treatments geared to optimize NOD2 signaling. 

There are limitations to the interpretation of our findings. While the adult patients included in this study are representative of cohorts included in multinational clinical trials, larger cohorts are needed to fully understand the range of immune responses that may be found in specific ethnic populations and/or age sub-groups. In addition to validated biomarkers in AD, we found a low but significant elevation of I-309/CCL1, known to recruit Th2 cells; the role of this cytokine in AD requires further investigation. Our in vitro model replicates effector AD-like type 2 T cell responses in blood, yet this only constitutes a fraction of the full spectrum of AD immune dysfunction found in the skin, where *S. aureus* phagocytosis and NOD2-mediated innate immune activation take place; stromal, endothelial, and epithelial cells are not featured. Chronic tissue-specific exposure to high burden *S. aureus* may functionally alter in situ immunity and T cell effector programs, emphasizing the need to study NOD2 response in human tissues. T lymphocytes in barrier tissues diverge significantly from populations found in the blood and exist in states that are more adaptive to microenvironmental factors, where NOD2-driven responses may vary, especially when considering tissue-resident T cells (Trm); this will be the subject of future dedicated studies. Interestingly, the topical application of a NOD2 de-repressor has recently been shown to induce the innate clearance of *S. aureus* in human skin explants; associated T cell responses remain to be investigated [88]. We focused most of our in vitro work on PBMCs from healthy subjects, replicating similar findings in blood from small numbers of AD patients, but we were not powered to adequately study differences between healthy and AD groups in vitro, including responsiveness to SEB, TSLP, or NOD2L. The detailed testing of dose-dependent responses and donor variability needs future work in larger studies. Future investigation will help clarify longitudinal responses and individual cytokine kinetics, along with the impact of TCR agonism potency, including (expected) quantitative but also qualitative differences in cytokines. For example, IL-22 was induced by TSLP/SEB in vitro only upon TCR re-stimulation (Figure 3 versus Figure 6). SEB alone has been shown to induce mixed Th2/Th1 and variable donor responses according to TCR repertoire and prior sensitization, the full characterization of which is beyond the scope of this study. *S. aureus* PGN O-acetylation is known to decrease inflammasome responses, yet direct impact on NOD2 has not yet been shown and, thus, is postulatory. PGN-derived signaling is complex, and its capacity to induce inflammation is context-dependent, including TLR synergy and variable chemical modifications of the PGN monomer muramyl dipeptide (MDP). In addition, the murine work with MDP found seemingly contradictory NOD2-induced Th2 responses, suggesting variation based on species, ligand sub-type, scavengers, co-factors, and/or tissue microenvironments [67]. As NOD2 signaling is intimately interconnected with TLR2 [92,93] and other PRRs and the tissue microenvironment, more detailed studies are needed with an expanded scope of ligands to elucidate how such sensors interact and signal in the context of atopic skin disease. The wide-ranging potential applications of NOD2 agonism via PGN, its monomer MDP, derivatives, and synthetic compounds constitute the basis of expansive current and global research. Future studies will help to unravel the spectrum of signaling and cellular players involved in human skin.

## 5. Conclusions

Human PBMCs exposure to TSLP/SEB in vitro can be used as a simple model for testing AD-like biomarkers and type 2 T cell responses. Our findings support emerging evidence for the role of NOD2 in regulating type 2 adaptive immune responses at the skin barrier surface in humans and highlight a novel endogenous mechanism modulating pathogenic IL-13 that may open an avenue for future translational research in AD. 

## Figures and Tables

**Figure 1 biomolecules-13-00369-f001:**
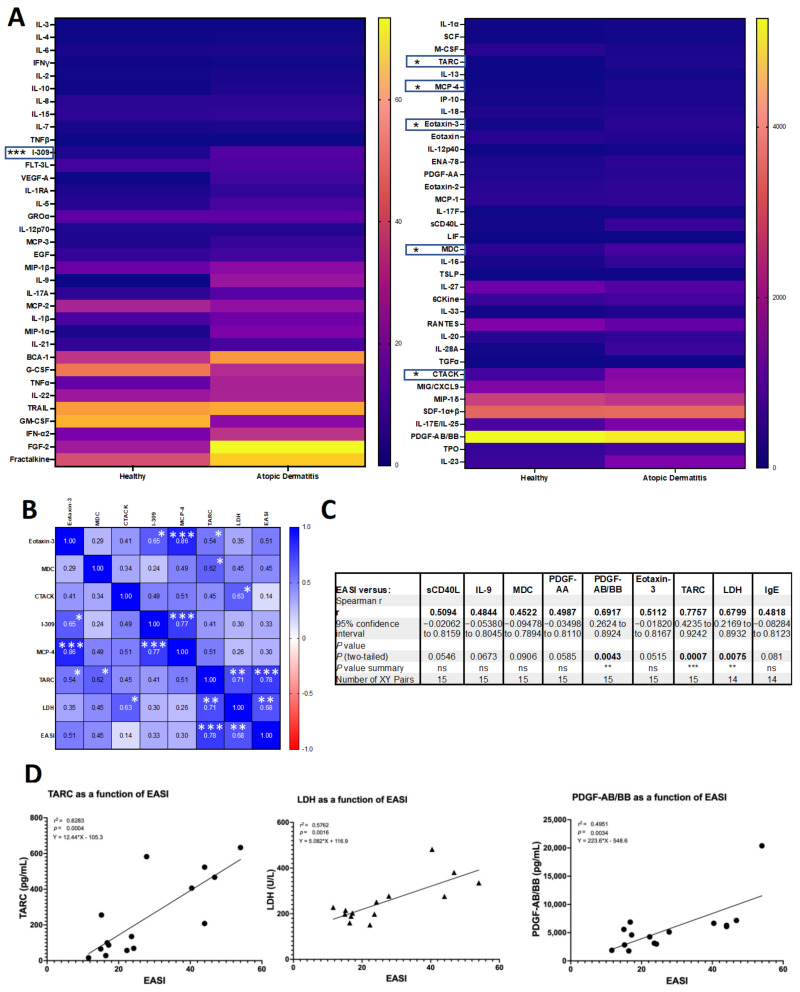
**Th2-related chemokines are significantly upregulated in atopic dermatitis patient blood and TARC significantly correlates with disease severity**. (**A**) Heatmap of median plasma cytokine and chemokine concentrations (pg/mL) determined using a 71-plex immunoassay; results are visualized according to concentration (0–50 pg/mL, left-panel; 0–5000 pg/mL, right-panel); moderate-to-severe adult AD patients (n = 15), matched healthy subjects (n = 7); statistics calculated using the non-parametric Mann-Whitney test per row, with individual ranks for each comparison followed by Holm-Šídák’s multiple comparisons test, alpha 0.05. Significantly elevated chemokines are indicated with asterisks; unadjusted *p* < 0.001, *p* values adjusted for multiple comparisons indicated. Not indicated are sCD40L, FGF-2, IL-13, IP-10, MCP-3, TFNα, VEGF-A, BCA-1 (unadjusted *p* < 0.05, all) and IL-9, IL-5, (*p* < 0.01, both) as significance is lost following adjustments for multiple comparisons. (**B**–**D**) AD patients (n = 15); (**B**) correlation between significantly elevated biomarkers, lactate dehydrogenase reported by clinical laboratory (U/L), and AD patient disease severity, measured according to the Eczema Area and Severity Index (EASI), indicating Spearman’s rho. (**C**) TARC, LDH, and PDGF-AB/BB correlate significantly with disease severity (EASI); analytes with a trend toward correlation are included for comparison (sCD40L, IL-9, MDC, PDGF-AA, Eotaxin-3, IgE). (**D**) Simple linear regression, modelling TARC chemokine, LDH, and PDGF-AB/BB growth factor concentrations (pg/mL) as a function of EASI. (**A**–**D**) * *p* ≤ 0.05, ** *p* ≤ 0.01, *** *p* ≤ 0.001. ns = not significant.

**Figure 2 biomolecules-13-00369-f002:**
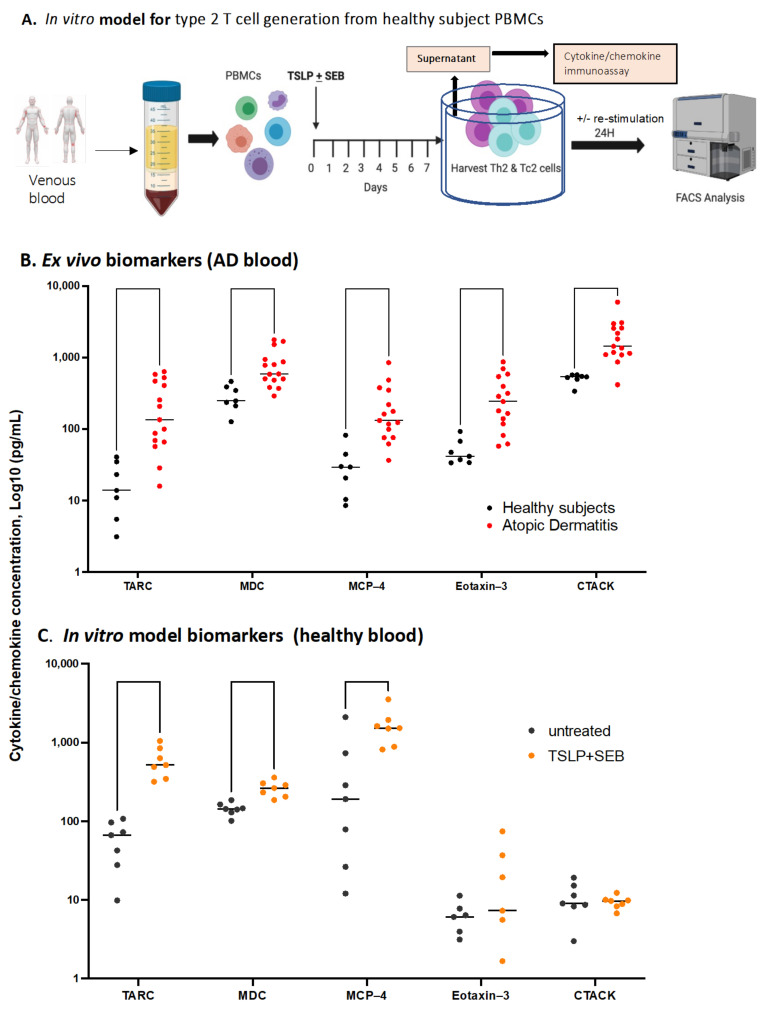
**The alarmin TSLP drives AD-like biomarker secretion from healthy PBMCs when combined with *S. aureus*-derived toxin SEB**. (**A**) Illustration of human in vitro type 2 T cell and cytokine enrichment and detection following culture with TSLP/SEB. (**B**) Five validated Th2-related AD chemokine biomarkers tested in plasma found significantly upregulated in the adult AD patient cohort (n = 15) relative to healthy subjects (n = 7), Mann-Whitney tests per row, followed by Holm-Šídák’s correction for multiple comparisons. Significant elevation for CTACK, unpaired *t*-test per row, followed by Holm-Šídák’s correction *(p* < 0.000001, data not shown). (**C**) In vitro stimulation with TSLP/SEB induces healthy PBMCs to secrete significantly elevated AD biomarkers following one week in culture (n = 7). Paired *t* tests per row, followed by Holm-Šídák’s multiple comparisons test. Eotaxin-3/CCL26 and CTACK/CCL27 are produced by cell-types (endothelia, epithelia) that are not found in PBMCs.

**Figure 3 biomolecules-13-00369-f003:**
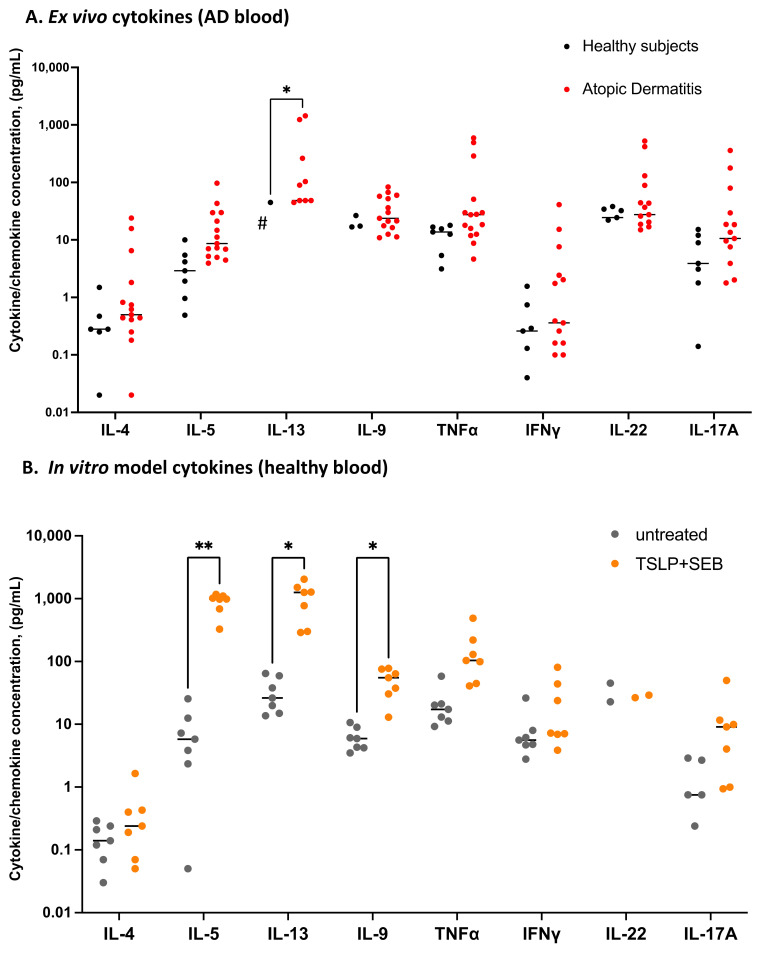
**Stimulation with TSLP/SEB in vitro induces AD-like type 2 cytokine secretion from healthy PBMCs.** Master cytokines secreted by type 1, 2, 3, or 22 polar T cells, as well as a regulatory (IL-10) cytokine, were tested ex vivo and in vitro. (**A**) Adult AD patients’ (n = 15) plasma tested ex vivo, relative to healthy subjects (n = 7). Significant IL-13 elevation, unpaired *t* tests per row, followed by Holm-Šídák’s multiple comparisons test. Significant for IL-5 by Mann-Whitney tests per row, followed by Holm-Šídák’s correction (*p* < 0.05, data not shown). (**B**) Stimulation with TSLP/SEB induces type 2 cytokines in vitro. PBMCs from healthy subjects (n = 7) cultured with TSLP/SEB one week, followed by collection of supernatants, with no mitogen re-stimulation; polar cytokine panel as in A. Paired *t* test per row, followed by Holm-Šídák’s multiple comparisons test. (**A**,**B**) * *p* ≤ 0.05, ** *p* ≤ 0.01. #, note: log scale; zero values not plotted on log scales.

**Figure 4 biomolecules-13-00369-f004:**
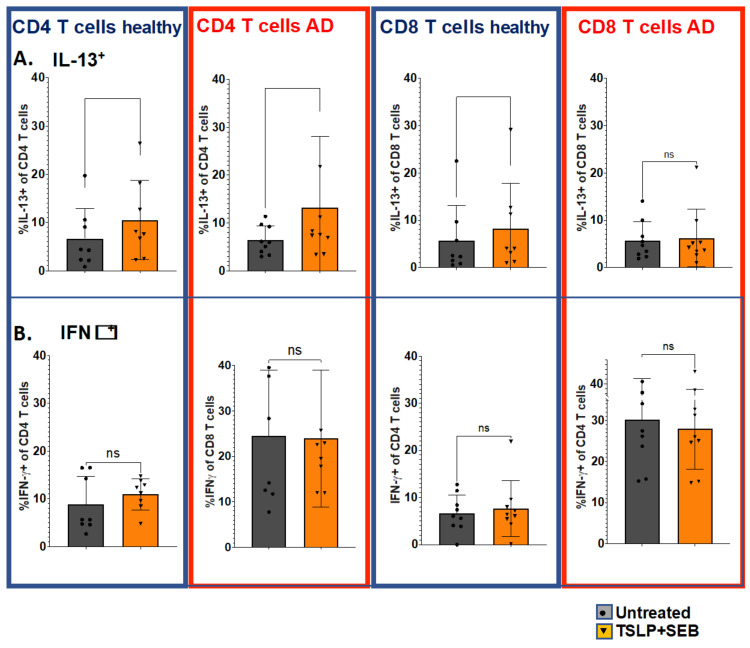
**In vitro stimulation with TSLP/SEB enriches IL-13^+^ CD4 and CD8 T cells from healthy PBMCs.** Intracellular cytokine detection by flow cytometry, following culture and re-stimulation (PMA/Ionomycin/GolgiStop, 6h) for intracellular cytokine detection ((**A**); IL-13, and (**B**); IFNγ). PBMCs from healthy subjects (n = 8) cultured with TSLP/SEB for one week; proportion (%) of IL-13^+^ (Th2/Tc2) versus IFNγ^+^ (Th1/Tc1) CD4/CD8 T cells. Statistical significance was determined by Wilcoxon matched-pairs signed rank test. Experiments performed in duplicates or triplicates, ns = non significant.

**Figure 5 biomolecules-13-00369-f005:**
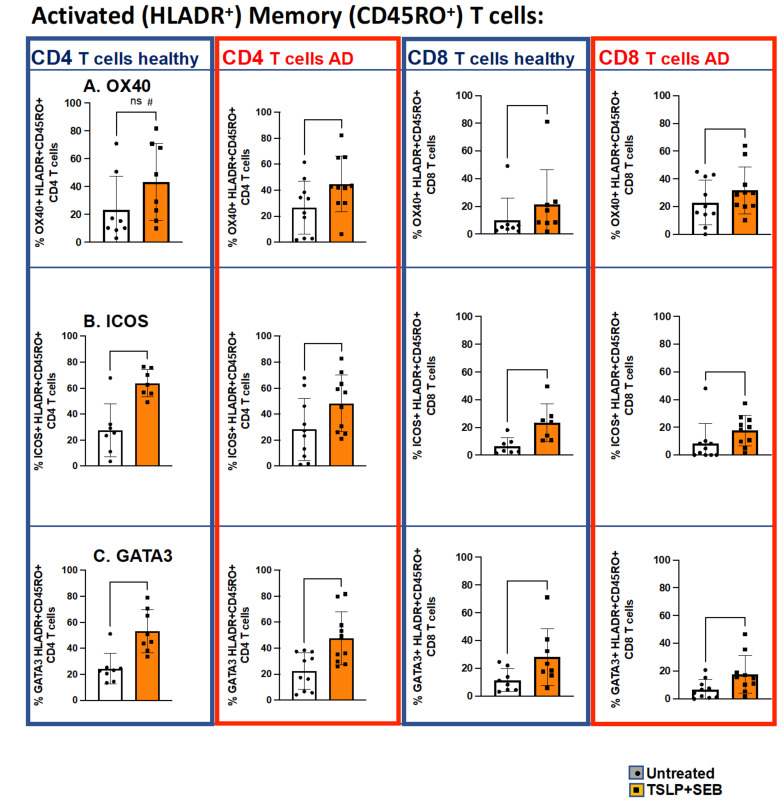
**In vitro TSLP/SEB significantly increases GATA3^+^, OX40^+^, and ICOS^+^ activated memory CD4 and CD8 T cells in PBMCs.** PBMCs from healthy subjects (n = 8, blue columns) or AD subjects (n = 10, red columns) were cultured with TSLP/SEB for one week, followed by multiparametric staining and flow cytometry, with no mitogen restimulation. Proportion (%) of co-stimulatory receptor (row A, OX40; row B, ICOS) and transcription factor (row C, GATA3)-expressing activated (HLADR^+^) memory (CD45RO^+^) CD4 (left-hand panels) and CD8 (right-hand panels) are indicated. Statistical significance was determined by Wilcoxon matched-pairs signed rank test. Experiments performed in duplicates or triplicates, # indicates a trend (*p* = 0.0781) toward increased % OX40 HLADR^+^CD45RO^+^CD4 T cells in treated healthy PBMCs. ns = non significant.

**Figure 6 biomolecules-13-00369-f006:**
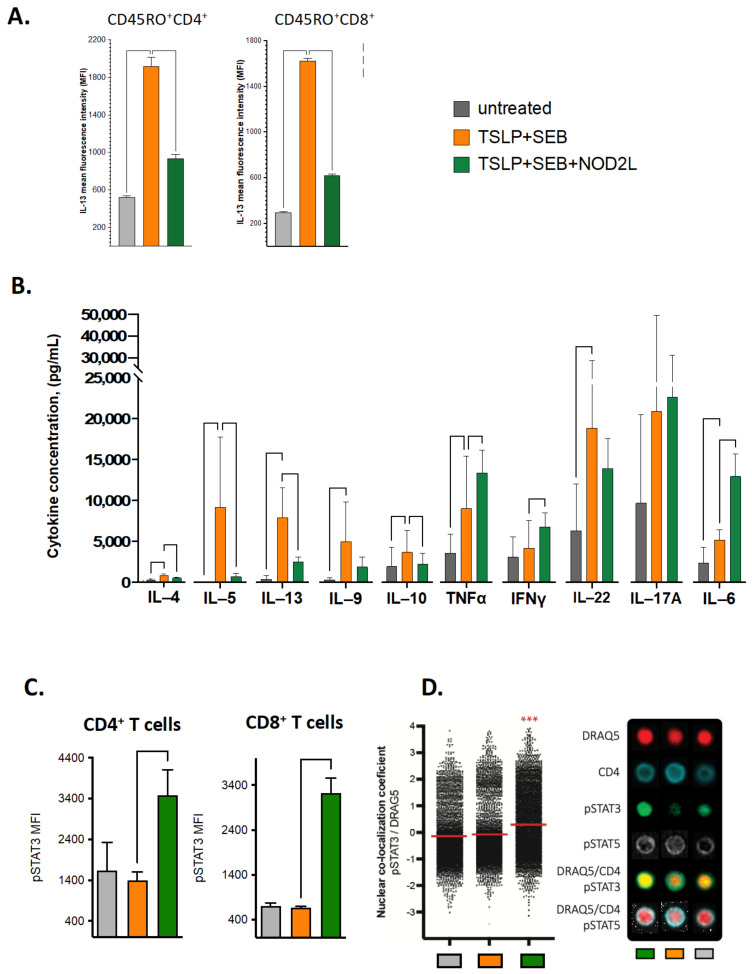
**NOD2 agonism inhibits TSLP/SEB-driven activation of Th2/Tc2 cell function in healthy PBMCs.** (**A**–**D**) In vitro stimulation of healthy subject PBMCs with TSLP/SEB, with or without NOD2-agonism for one week; cells were then re-stimulated with anti-CD3/CD28 beads for 6h prior to supernatant collection for cytokine detection and/or staining of cells for flow cytometry analysis, experiments performed in duplicates (n = 3). (**A**) Memory CD4 and CD8 T cell (CD45RO^+^) IL-13 secretion, mean fluorescence intensity (MFI) detected using cell-surface capture sandwich secretion assay, statistical analysis by one-way ANOVA, Dunnett’s multiple comparison’s test, relative to control (TSLP/SEB + Vehicle). (**B**) NOD2-agonism significantly decreased Th2 (IL-4, IL-5, IL-13) and IL-10 production and significantly increased IL-6, TNF, and IFNγ; statistical analysis by one-way ANOVA was calculated for each cytokine and corrected with Dunnett’s multiple comparisons test, *p*-values as in the post-hoc test. (**C**) NOD2-agonism significantly increased phosphorylated (p)STAT3 expression in CD4 and CD8 T cells, mean fluorescence intensity (MFI). (**D**) Significant increase in pSTAT3 co-localization with nuclear factor (DRAQ5) in CD4 T cells; left, indicating mean (read), one-way ANOVA followed by paired *t* test comparing TSLP/SEB +/− NOD2L; right, representative imaging of co-localized pSTAT3, versus pSTAT5. *** *p* < 0.001.

## Data Availability

The data that support the findings of this study are included in the manuscript and Appendix A.

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
