# Peer review of "NOD2 Agonism Counter-Regulates Human Type 2 T Cell Functions in Peripheral Blood Mononuclear Cell Cultures: Implications for Atopic Dermatitis"

_biomolecules, 2023, doi:10.3390/biom13020369_

Round 1

Reviewer 1 Report

This manuscript is well written and interesting.

It will be accepted after some correction.

- In Fig. 4., I can not find a mark "A", and "B" .

- What is the possible molecular signaling mechanism of NOD2 action for Th2 inhibition besides pSTAT3 ?

- How do you expect the results to change if you substitute the representative allergic inducer such as albumin or house dust mites, instead of SEB? A brief discussion of the effects of different stimuli is recommended.

Reviewer 2 Report

Using human samples, Gimenez-Rivera et al. describe the effects of the S. aureus superantigen SEB and counteracting effect of the nucleotide-binding oligomerization domain-containing protein 2 (NOD2) agonist on type 2 immunity.

Based on the observation that serum levels of thymus and activation-related chemokine (TARC)/CCL17, lactate dehydrogenase (LDH), and platelet-derived growth factor (PDGF) were correlated with the severity of atopic dermatitis (AD). Next, they showed that healthy peripheral blood stimulation with thymic stromal lymphopoietin (TSLP)/SEB mimics AD peripheral blood phenotypes regarding chemoattractant/cytokine secretion. They also found that CD8 T cells from AD patients exhibited decreased interleukin 13 (IL-13) levels, whereas those in CD4 T cells were comparable. The finding was corroborated by preferentially upregulated OX40, a costimulatory molecule on memory T cells, in AD CD4 T cells. In contrast, those in another costimulatory molecule ICOS or type 2 signature transcription factor GATA3 were comparable. Finally, they showed a counteracting effect of a NOD2 agonist (ultra-purified S. aureus peptidoglycan) in the TSLP/SEB assay, presumably mediated through the activation of STAT3, a critical regulator of T helper (TH) 17 differentiation.

Because the TH2 and the TH17 oppose each other, this manuscript described straightforward results and was comfortable to read. The strength of this study is that they used human samples from the standpoint of translational research.

Here are the comments.

Major

The results all look somewhat within expectation, and they need to describe the novelty of this study. 

Another drawback of this study would be that they ended up using the peripheral blood but not skin samples; as in L 73, normal human skin has abundant S. aureus-specific tissue-resident memory CD4 T cells that produce IL-17A/IFN-γ. Moreover, given that S. aureus phagocytosis and NOD2-mediated innate immune activation take place in the skin but not the peripheral blood (ref. 41); therefore, the authors have to describe this crucial issue in the limitation statement.

Minor

1. Figure 6D, right panel: the authors should present the picture, including the TSLP/SEB + Vehicle (orange) group.

Reviewer 3 Report

The authors have performed an interesting study that revealed the role of NOD2 in the pathogenesis of atopic dermatitis. However there are several issues that should be addressed.

For the abbreviation of thymic stromal lymphopoetin the authors used TLSP and TSLP. Please revise it.

The last paragraph in the Introduction Section represents the conclusions of the study. This paragraph should be replaced with a paragraph in which the aim and objectives of the study are clearly presented.   The authors mentioned the number of study participants in the Results section. However I consider these data should be included in the Materials and methods section.

Lines 109-110 “Patients meeting parallel inclusion criteria were recruited from Innovaderm REB 6036 for mechanistic work with fresh cells in vitro.” What do the authors mean when they say meeting parallel inclusion criteria?

The authors should provide more data about Innovaderm in the Study design Subsection.

Line 136 "Commercial NOD2 agonist (NOD2L), ultrapurified S. aureus peptigoglycan (Invivogen)." - Please rephrase it.

I consider that the tests used for the statistical analysis should be mentioned in the Statistical analysis subsection.

In the Results section the authors should present only the results obtained and not include comments on the results or comparisons with other studies. It is difficult to read the results in this form. Comments like the ones below should be moved to the Discussion section.

Lines 281-284 "In summary, we found that this in vitro TSLP/SEB polarization model effectively replicates type 2 T cell functional deviation seen in AD patients by leveraging two key disease co-factors known to play a strong role in the skin 283 lesions of patients."

Lines 310-311 "We also note the lack of IL-10 production, in contrast to prior reports with cell wall extracts of S. aureus or TLR2 311 ligation [54, 55]."

The Keywords section seems to be a list of abbreviations. Please revise it.

The authors should provide a title and a legend for each figure.

The authors should revise the numbering of the subsections.

Round 2

Reviewer 3 Report

The manuscript has been significantly improved.